# Assessment of Sea-Surface Wind Retrieval from C-Band Miniaturized SAR Imagery

**DOI:** 10.3390/s23146313

**Published:** 2023-07-11

**Authors:** Yan Wang, Yan Li, Yanshuang Xie, Guomei Wei, Zhigang He, Xupu Geng, Shaoping Shang

**Affiliations:** 1College of Ocean and Earth Sciences, Xiamen University, Xiamen 361005, China; 22320210156154@stu.xmu.edu.cn (Y.W.);; 2Key Laboratory of Underwater Acoustic Communication and Marine Information Technology, Minister of Education, Xiamen University, Xiamen 361005, China; 3Dongshan Swire Marine Station, Xiamen University, Xiamen 361005, China; 4Engineering Research Center of Ocean Remote Sensing Big Data, Fujian Province University, Xiamen 361102, China; 5State Key Laboratory of Marine Environmental Science, Xiamen University, Xiamen 361005, China

**Keywords:** HiSea-1, Chaohu-1, wind, SAR

## Abstract

Synthetic aperture radar (SAR) has been widely used for observing sea-surface wind fields (SSWFs), with many scholars having evaluated the performance of SAR in SSWF retrieval. Due to the large systems and high costs of traditional SAR, a tendency towards the development of smaller and more cost-effective SAR systems has emerged. However, to date, there has been no evaluation of the SSWF retrieval performance of miniaturized SAR systems. This study utilized 1053 HiSea-1 and Chaohu-1 miniaturized SAR images covering the Southeast China Sea to retrieve SSWFs. After a quality control procedure, the retrieved winds were subsequently compared with ERA5, buoy, and ASCAT data. The retrieved wind speeds demonstrated root mean square errors (RMSEs) of 2.42 m/s, 1.64 m/s, and 3.29 m/s, respectively, while the mean bias errors (MBEs) were found to be −0.44 m/s, 1.08 m/s, and −1.65 m/s, respectively. Furthermore, the retrieved wind directions exhibited RMSEs of 11.5°, 36.8°, and 41.7°, with corresponding MBEs of −1.3°, 2.4°, and −8.8°, respectively. The results indicate that HiSea-1 and Chaohu-1 SAR satellites have the potential and practicality for SSWF retrieval, validating the technical indicators and performance requirements implemented during the satellites’ design phase.

## 1. Introduction

Seasat, launched by the United States in 1978, represents one of the initial ocean observation satellites in human history and marked the initiation of ocean observation from space [1]. The development of synthetic aperture radar (SAR) technology has paved the way for the successful development and launch of numerous SAR satellites, including ERS-1/2, RADARSAT-1/2, ENVISAT, Sentinel-1, and Gaofen-3. These C-band SARs have found widespread use in observing specific ocean parameters, such as sea-surface wind fields (SSWFs), wave fields, and sea ice [2,3,4,5,6,7,8,9,10,11,12], and some scholars have evaluated the SSWF retrieval capabilities of several SAR sensors [13,14,15,16,17,18]. Nevertheless, traditional SAR satellites are large and costly to construct and launch, resulting in expensive SAR data, which constitutes one of the factors that restrict the widespread application of SAR imagery. Therefore, SAR satellites are progressively moving towards miniaturization and lower cost. At present, the C-band miniaturized SAR satellites that have been successfully launched internationally are HiSea-1 and Chaohu-1.

HiSea-1 is China’s first commercial miniaturized C-band synthetic aperture radar (SAR) satellite. This SAR satellite was developed in collaboration with Xiamen University, the China Electronics Technology Group Corporation (CETC) No.38 Institute, and Spacety, and successfully launched on 22 December 2020. With its high-resolution, all-weather, and cloud-penetrating capabilities, HiSea-1 can provide precise observation of oceans, land, and polar regions. The satellite uses a single VV (vertical transmit and vertical receive) polarization and has a total mass that does not exceed 185 kg. HiSea-1’s data products comprise Level-1 single look complex (SLC) and Level-2 orthorectification geolocation (ORG) images, and its imaging modes include stripmap, ScanSAR (narrow ScanSAR/extra ScanSAR), and spotlight. Its spatial resolution can reach up to 1 m, and its swath width can extend up to 100 km [19]. With global coverage prospects, the satellite offers extensive application possibilities, including marine environment monitoring, resource investigation, and disaster monitoring. Specifically, it can monitor SSWFs, wave heights, ocean currents, and oil spills on the sea surface, thereby rendering effective support to the preservation of the marine environment. Chaohu-1, launched on 27 February 2022, is a SAR satellite jointly developed by the Institute of Space Integrated Ground Network (Anhui), the CETC No.38 Institute, and Spacety. Chaohu-1 represents China’s second commercial C-band miniaturized SAR satellite after HiSea-1 and the first launch of the “Tianxian” constellation project in China. The satellite possesses parameters that are similar to those of HiSea-1; however, its platform and radar payload design have undergone further optimization, resulting in significant improvements in core capabilities such as imaging swath, resolution, maximum imaging duration, data transmission, and orbit control. Furthermore, Chaohu-1 has added the capability of continuous imaging of multipoint targets in the region, precise orbit determination, and on-orbit artificial intelligence processing functionality.

HiSea-1 and Chaohu-1 both possess spatial resolutions of up to 1 m, providing precise and high-resolution SSWFs of the ocean surface, particularly in coastal regions. Presently, the geophysical model function (GMF) is the most extensively employed approach for SSWF retrieval from SAR images. For C-band VV-polarization SAR images, commonly employed GMFs encompass CMOD4 [20], CMOD_IFR2 [21], CMOD5 [22], CMOD5.N [23], and CMOD7 [24].

At present, two methods are available for SSWF retrieval from SAR images using the CMOD: the direct retrieval method [25,26] and the variational method [10,27,28,29]. The direct retrieval method utilizes wind direction as prior information, which is input into the CMOD to obtain wind speed. The prior wind direction information primarily originates from wind streak information within SAR images themselves [30,31,32], or from external SSWF data. However, the wind direction extracted from the wind streaks is 180° ambiguous, and wind streak information is not widely distributed in SAR images. Hence, in practical applications, external SSWF data such as numerical weather prediction, ECMWF reanalysis v5 (ERA5), and cross-calibrated multiplatform (CCMP), are more frequently used. A significant drawback of the direct retrieval method is that errors in the prior wind direction can affect the accuracy of the retrieved wind speed. Unlike the direct inversion method, the variational method combines the GMF, normalized radar cross-section (NRCS, σ_0_), and external SSWF data to construct a cost function. By obtaining the minimum value of the cost function, the optimal wind direction and wind speed are determined. Previous research indicates that the variational method achieves higher accuracy in SSWF retrieval than the direct retrieval method [28].

In this study, we conducted a preliminary assessment of SSWFs retrieval from HiSea-1 and Chaohu-1 SAR images. Specifically, the lookup table computed by CMOD5.N and CCMP wind vectors was input into the cost function to obtain the final SSWF retrieval results. The remainder of this paper is organized as follows. Section 2 presents the data utilized in this study, which include HiSea-1 and Chaohu-1 SAR images, buoy winds, ASCAT winds, and ERA5 winds. Section 3 shows the SSWF retrieval method. Section 4 shows the quality control procedure. Subsequently, the SSWF retrieval results and discussion are shown in Section 5. Conclusions are summarized in Section 6.

## 2. Data

### 2.1. HiSea-1 and Chaohu-1 SAR Images

In this study, we collected 1053 SAR images, including 378 HiSea-1 SAR images and 675 Chaohu-1 SAR images, to retrieve SSWFs. These SAR images covered the Southeast China Sea area and were acquired in stripmap mode with a product type of Level-2 ORG. Each SAR image covered an area of approximately 25 km × 25 km with 3 m resolution. The imaging time ranged from March 2022 to December 2022. Figure 1 illustrates the spatial coverage of these images.

### 2.2. Buoy Winds

In this study, buoy winds from 16 stations were collected to evaluate the SSWFs retrieved from the 1053 SAR images. The locations of these buoy stations are depicted in Figure 1, where blue stars represent the buoys recording 10 m winds, while blue triangles represent the buoys recording 3 m winds. A conversion formula is required to obtain the 10 m wind speeds from the 3 m wind speeds. The conversion formula is provided below [33]:(1)$$\frac{U_{10}}{U_{3}}=\frac{\ln(10/z_{0})}{\ln(3/z_{0})}$$
wherein $U_{10}$ and $U_{3}$ represent the wind speed at 10 m height and 3 m height, respectively. The constant *z*_0_ represents the roughness length. The value of *z*_0_ depends on the nature of the terrain and must be determined empirically [34], and the typical value is 1.52 × 10^−4^ [18,33,34].

### 2.3. CCMP Winds, ERA5 Winds, and ASCAT Winds

In this study, CCMP data were employed to facilitate the retrieval of SSWFs from SAR images, while ASCAT and ERA5 data were utilized for the assessment of the retrieved wind speeds. The CCMP data have a spatial resolution of 0.25° × 0.25° and a temporal resolution of 6 h, whereas ERA5 data have a spatial resolution of 0.25° × 0.25° and a temporal resolution of 1 h. The ASCAT orbit data or ASCAT swath product were also utilized in this study.

The comparison results between the buoy-measured 10 m wind speeds and ASCAT, ERA5, and CCMP wind speeds in the Southeast China Sea are illustrated in Figure 2. It is evident from the figure that CCMP wind speeds exhibit the highest accuracy, with a root mean square error (RMSE) of 1.63 m/s. ERA5 wind speeds demonstrate a slightly lower accuracy of 1.91 m/s, but with an underestimation at high wind speeds. ASCAT wind speeds display the lowest accuracy, with an RMSE of 2.47 m/s. In other words, the errors between the three datasets and buoy-measured wind speeds were acceptable.

### 2.4. Data for Quality Control

Additionally, 720 Sentinel-1 ground range detected (GRD) images were utilized to retrieve SSWFs via the variational method in this study. These SAR images were acquired in the interferometric wide (IW) swath mode and covered the Southeast China Sea.

In general, wind speed and mean square slope of the sea surface have a good linear correlation [35], and outliers of retrieved SSWFs can be identified and removed based on this relationship. However, the $\sigma_{0}$ of SAR imagery is influenced by intricate factors such as oceanic and atmospheric parameters, as well as sea–air interactions. Thus, the relationship between the wind speed and the $\sigma_{0}$ cannot be viewed as a simple linear relationship. In this study, to ensure the quality of the SSWF retrieval from HiSea-1 and Chaohu-1 SAR images, an empirical quality-controlled region was established based on the retrieved wind speeds and $\sigma_{0}$ from Sentinel-1.

## 3. Method and Results

So far, GMF has been widely employed for SAR SSWF retrieval. The GMF’s general form is as follows:(2)$$\sigma_{0}^{sim}=B_{0}(U,\theta ){\left[1+B_{1}(U,\theta )\cos(\phi )+B_{2}(U,\theta )\cos(2\phi )\right]}^{\gamma }$$
wherein $\sigma_{0}^{sim}$ denotes the NRCS simulated by GMF, *U* denotes the wind speed, *θ* denotes the incidence angle, ϕ denotes the angle between the wind direction and the radar look angle, and *γ* is a constant. This study employs the variational method, coupled with CMOD5.N and CCMP reanalysis datasets, to estimate the SSWF from 1053 HiSea-1 and Chaohu-1 SAR images. The retrieval proceeds through a three-step procedure.

In Step 1, the SAR imagery is partitioned into multiple wind cells, each possessing a coverage of 1 km × 1 km. Hence, the spatial resolution of the retrieved SSWFs is 1 km × 1 km. Additionally, the CCMP winds are interpolated from the original 0.25° × 0.25° resolution to 1 km × 1 km.

In Step 2, a homogeneity check on the normalized radar cross-section (NRCS) distribution of each wind cell is performed via the normalized variance. The normalized variance is defined using the following formula:(3)$$A=\mathrm{var}(\frac{\sigma_{0}-{\bar{\sigma }}_{0}}{{\bar{\sigma }}_{0}})$$
where *A* represents the normalized variance and var(•) represents the operator for calculating the variance of a dataset. $\sigma_{0}$ represents the NRCS of a wind cell, whereas ${\bar{\sigma }}_{0}$ represents the average NRCS of a wind cell. In this study, wind cells are included in the SSWF retrieval calculation when the value of *A* is less than 1.8, indicating a homogeneous NRCS distribution within the cell. Conversely, wind cells with a value of *A* greater than or equal to 1.8 are excluded from the SSWF retrieval calculation.

In Step 3, the SSWFs are retrieved utilizing the variational method. The mathematical formulation for this method is expressed in terms of a variational formula (or cost function), as represented by Equation (4):(4)$$J=(\frac{u-u^{r}}{\Delta u})+(\frac{v-v^{r}}{\Delta v})+(\frac{{\bar{\sigma }}_{0}-\sigma_{0}^{sim}}{\Delta \sigma })$$
wherein *u* and *v* represent the eastward and northward components of the retrieved SSWF, respectively, and $u^{r}$ and $v^{r}$ denote the corresponding components of the CCMP SSWF. The variable ${\bar{\sigma }}_{0}$ represents the average NRCS of a wind cell (the observed value of SAR), and $\sigma_{0}^{sim}$ represents the simulation value of CMOD5.N. Here, Δu=2 m/s, Δv=2 m/s, and Δσ=0.1 dB [29].

Figure 3a,b show the SSWFs retrieved from the HiSea-1 SAR image in stripmap mode at 15:56 UTC on 8 June 2022 and 04:05 UTC on 7 April 2022, respectively. Figure 3c,d show the SSWFs retrieved from the Chaohu-1 SAR image in stripmap mode at 02:55 UTC on 8 December 2022 and 14:34 UTC on 22 December 2022, respectively. Figure 3e shows the SSWFs retrieved from the HiSea-1 SAR image in stripmap mode at 15:26 UTC on 15 July 2022 and 14:34 UTC on 22 December 2022. As shown in Figure 3a, the beach and the nearby underwater topography can be clearly observed, as well as the variations in the details of the wind field. In Figure 3b–d, significant differences in wind speeds are observed on both sides of the frontal surface, with the largest difference seen in Figure 3c, reaching around 5 m/s. Figure 3e shows the changing wind direction.

## 4. Quality Control Procedure

In this study, 720 Sentinel-1 SAR images were utilized to retrieve SSWFs, and the retrieval method is described in Section 3. Owing to HiSea-1 and Chaohu-1 SAR images with a coverage of 25 km × 25 km, each Sentinel-1 SAR image was partitioned into 25 km × 25 km subimages. The average wind speed and $\sigma_{0}$ variance of each subimage were calculated to establish an empirical quality-controlled region. As illustrated in Figure 4, the quality-controlled region takes the form of an elliptical region and is expressed by Equation (5).
(5)$$\frac{{\left(x\cos\beta -y\sin\beta \right)}^{2}}{a^{2}}+\frac{{\left(x\sin\beta +y\cos\beta \right)}^{2}}{b^{2}}=1$$
where *a* = 2, *b* = 0.35, and β=83°. We expected the relationship between the average retrieved wind speeds and $\sigma_{0}$ variances obtained from HiSea-1 and Chaohu-1 SAR images to match those obtained from Sentinel-1 SAR images. Consequently, the HiSea-1 and Chaohu-1 SAR images that derived data points outside the quality-controlled region were removed.

## 5. Discussion

Following the quality control in Section 4, a comparison was conducted between the wind speeds retrieved from HiSea-1 and Chaohu-1 SAR images and those from ERA5, buoy, and ASCAT. Figure 5 depicts the comparison results.

Due to the HiSea-1 and Chaohu-1 SAR images with a coverage of 25 km × 25 km, almost every SAR image covers one ERA5 grid point. Consequently, the wind vectors retrieved from each SAR image were averaged, and the obtained average was compared with the wind vector at the ERA5 grid point contained in the image. As illustrated in Figure 5a,b, the SAR-retrieved wind speeds exhibit an RMSE of 2.42 m/s and a mean bias error (MBE) of −0.44 m/s; the SAR-retrieved wind directions demonstrate an RMSE of 11.5° and an MBE of –1.3° when compared with ERA5 winds.

As a result of the limited swath of HiSea-1 and Chaohu-1 SAR images, and the absence of buoy data in December 2022, only 5 SAR images out of 1053 matched in both time and space with buoy data. The wind vectors were extracted from the wind cells covering the buoy locations in the five SAR images and compared with the corresponding buoy wind vectors. As presented in Figure 5c,d, the SAR-retrieved wind speeds reveal an RMSE of 1.64 m/s and an MBE of 1.08 m/s; the SAR-retrieved wind directions display an RMSE of 36.8° and an MBE of 2.4° when compared with buoy winds.

The selection of ASCAT orbit data was governed by two criteria: firstly, the data had to encompass the study area depicted in Figure 1, and secondly, the temporal discrepancy between ASCAT overpass and SAR image acquisition was constrained to be less than 60 min. The wind vectors were extracted from the wind cells covering the ASCAT data points in the SAR images and compared with the corresponding ASCAT wind vectors. As depicted in Figure 5e,f, the SAR-retrieved wind speeds exhibit an RMSE of 3.29 m/s and an MBE of −1.65 m/s; the SAR-retrieved wind directions exhibit an RMSE of 41.7° and an MBE of −8.8° when compared with ASCAT winds.

Based on these findings, the errors of the retrieved wind vectors from HiSea-1 and Chaohu-1 SAR images are acceptable.

## 6. Conclusions

The high costs of traditional large SAR systems have prompted a move towards smaller, more cost-effective SAR satellites. HiSea-1 and Chaohu-1 are examples of successfully launched miniaturized C-band SAR satellites. HiSea-1 is China’s first commercial miniaturized SAR satellite. The satellite employs a single-polarization VV mode and has a total mass not exceeding 185 kg. The SAR system can operate in three different imaging modes, with the spatial resolution reaching up to 1 m. Meanwhile, Chaohu-1 is the second miniaturized SAR satellite, which shares many parameters with HiSea-1 but features further optimizations in satellite platform and radar payload design. In contrast to previous SAR systems, HiSea-1 and Chaohu-1 exhibit comparable imaging performance despite their reduced size and weight. As a comparison, Gaofen-3 SAR satellite’s weight is 2750 kg and its cost of power is 15 kW, while HiSea-1 SAR satellite’s weight is 185 kg and its cost of power is 2.5 kW [19]. To summarize, this study aimed at investigating the potential of the two miniaturized SAR systems in SSWF retrieval.

The SSWF plays a pivotal role in influencing ocean circulation and is also the primary driving force for the generation of ocean waves. Amongst many studies pertaining to oceanography and meteorology, the SSWF stands as a crucial parameter. The HiSea-1 and Chaohu-1 satellites can provide high-resolution SSWF information. In this study, we employed the variational method and CMOD5.N to retrieve the SSWFs from 1053 HiSea-1 and Chaohu-1 SAR images. After a quality control procedure, the retrieved wind SSWFs were subsequently compared with ERA5, buoy, and ASCAT data. Compared to ERA5 winds, the retrieved wind speeds demonstrated an RMSE of 2.42 m/s and an MBE of −0.44 m/s; the retrieved wind directions demonstrated an RMSE of 11.5° and an MBE of −1.3°. In comparison to buoy winds, the retrieved wind speeds displayed an RMSE of 1.64 m/s and an MBE of 1.08 m/s; the retrieved wind directions displayed an RMSE of 36.8° and an MBE of 2.4°. Compared to ASCAT winds, the retrieved wind speeds demonstrated an RMSE of 3.29 m/s and an MBE of −1.65 m/s; the retrieved wind directions exhibited an RMSE of 41.7° and an MBE of −8.8°.

Drawing on the findings outlined above, we may infer that the errors in SSWFs retrieved from HiSea-1 and Chaohu-1 SAR images are acceptable. This attests to the potential and utility of these SAR satellites in SSWF retrieval and demonstrates the successful realization of the technical specifications and performance criteria established for these platforms during their design phase.

## Figures and Tables

**Figure 1 sensors-23-06313-f001:**
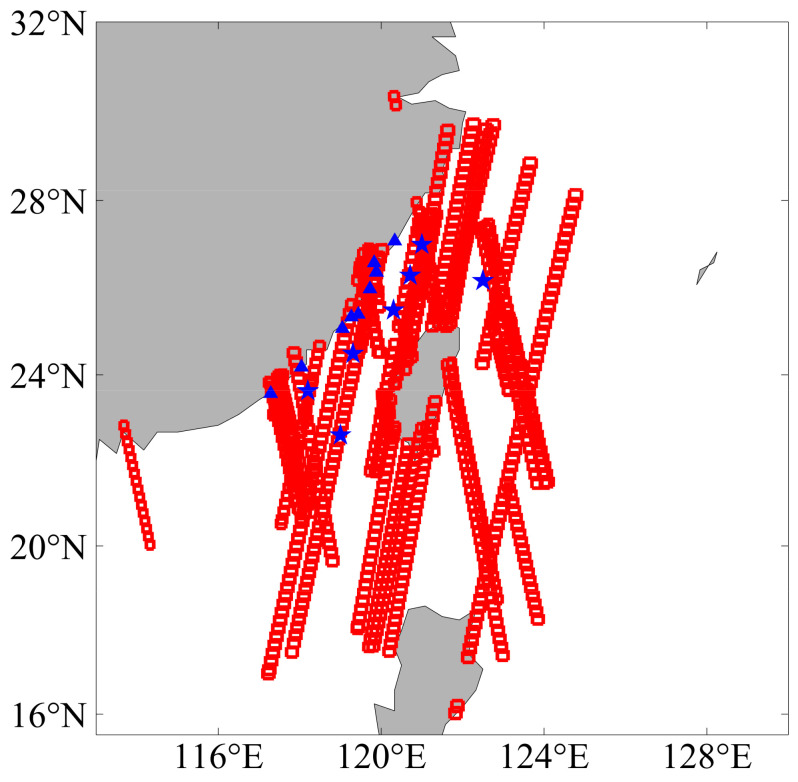
Coverage area of SAR images and locations of buoys. The red box outlines the coverage area of each SAR image, while blue stars and triangles represent the locations of the buoys recording winds at 10 m and 3 m height, respectively.

**Figure 2 sensors-23-06313-f002:**
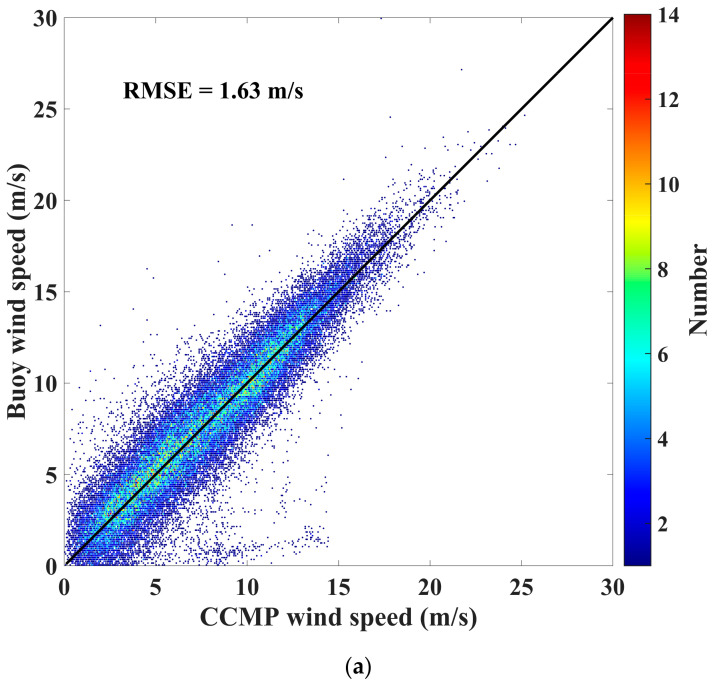
Buoy wind speeds versus (**a**) CCMP wind speeds, (**b**) ERA5 wind speeds, and (**c**) ASCAT wind speeds. The bisector is depicted by the black solid line.

**Figure 3 sensors-23-06313-f003:**
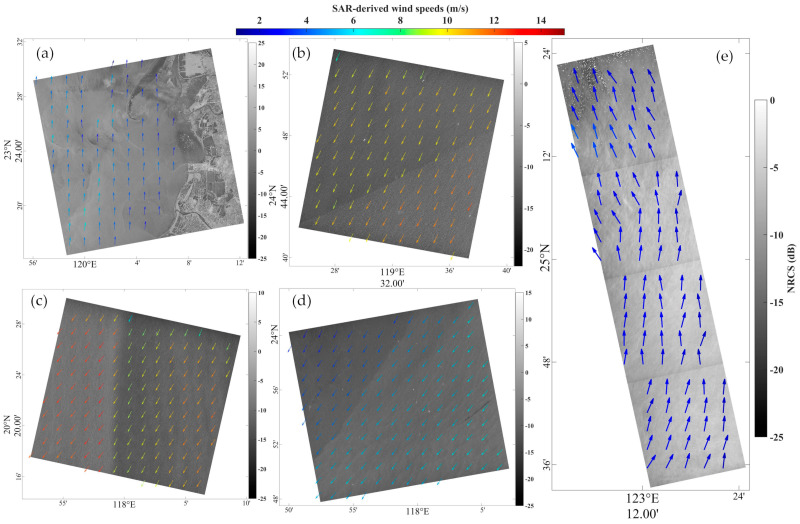
SSWFs retrieved from SAR images acquired by (**a**) HiSea-1 at 15:56 UTC on 8 June 2022, (**b**) HiSea-1 at 04:05 UTC on 7 April 2022, (**c**) Chaohu-1 at 02:55 UTC on 8 December 2022, (**d**) Chaohu-1 at 14:34 UTC on 22 December 2022, and (**e**) HiSea-1 at 15:26 UTC on 15 July 2022.

**Figure 4 sensors-23-06313-f004:**
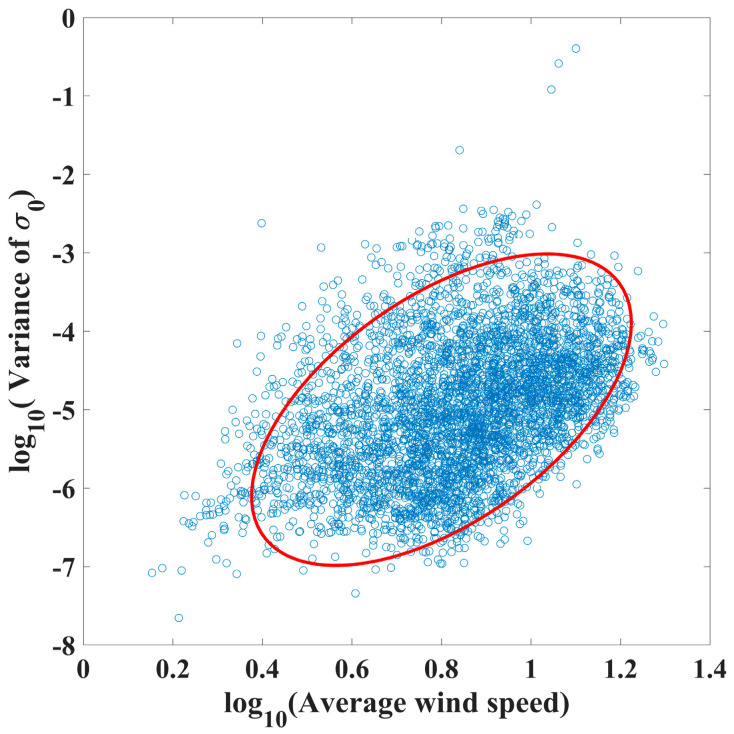
Variance versus average retrieved wind speed for each Sentinel-1 sub-SAR image with coverage of 25 km × 25 km. The red elliptic curve represents the quality-controlled region and is expressed by Equation (5).

**Figure 5 sensors-23-06313-f005:**
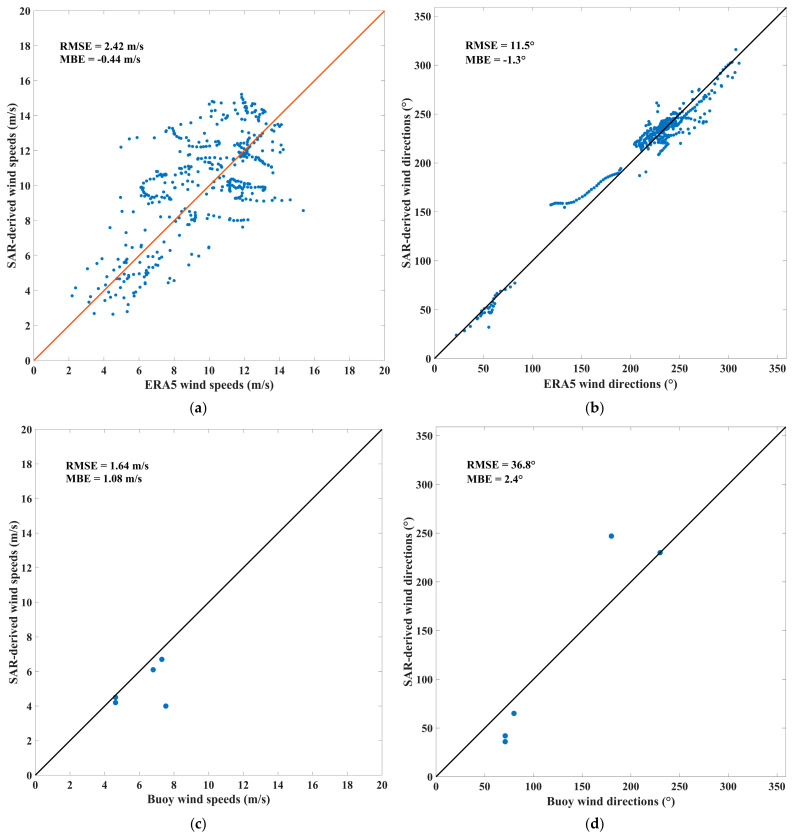
SAR-derived winds versus (**a**,**b**) ERA5 winds, (**c**,**d**) buoy winds, and (**e**,**f**) ASCAT winds.

## Data Availability

The ERA5, CCMP, and ASCAT datasets are downloaded from https://cds.climate.copernicus.eu/cdsapp#!/dataset/reanalysis-era5-single-levels?tab=form, https://data.remss.com/ccmp/v02.1.NRT/, and https://data.remss.com/ascat/. The buoy data are provided by Fujian Provincial Marine Forecast Station (www.fjhyyb.cn). The Sentinel-1 SAR data can be downloaded via https://search.asf.alaska.edu/. The authors would also like to thank the HiSea-1 C-band SAR satellite project and all collaborators, especially Fujian Tendering Purchasing Group Co., Ltd., Fuzhou, China and Fujian Haisi Digital Technology Co., Ltd., Sanming, China.

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
