# Peer review of "Assessment of Sea-Surface Wind Retrieval from C-Band Miniaturized SAR Imagery"

_sensors, 2023, doi:10.3390/s23146313_

Round 1
Reviewer 1 Report
The article describes the work of two Chinese satellites HiSea-1 and Chaohu-1 and analyzes observing sea surface wind fields. The main idea of the paper is to evaluate the sea surface wind field retrieval performance of miniaturized SAR systems in the Southeast China Sea. The authors compare the data from Chinese satellites HiSea-1 and Chaohu-1 with ERA5, buoy, and ASCAT wind speeds, respectively. The authors demonstrate the potential and utility of these SAR satellites in the sea surface wind fields retrieval and the successful realization of the technical specifications and performance criteria established for these platforms.
There are some comments:
1) What does the transformation formula (1) follow from? It is necessary to give relevant links or justify the transformation formula (1).
2) For what reasons is z0=… chosen? Justifications are required.
3) Missing formula (2).
4) The text is a bit boring.

Author Response
Reviewer1: The article describes the work of two Chinese satellites HiSea-1 and Chaohu-1 and analyzes observing sea surface wind fields. The main idea of the paper is to evaluate the sea surface wind field retrieval performance of miniaturized SAR systems in the Southeast China Sea. The authors compare the data from Chinese satellites HiSea-1 and Chaohu-1 with ERA5, buoy, and ASCAT wind speeds, respectively. The authors demonstrate the potential and utility of these SAR satellites in the sea surface wind fields retrieval and the successful realization of the technical specifications and performance criteria established for these platforms.
There are some comments:
Response: Thanks for the positive comments. All of your comments are addressed below in detail.
1) What does the transformation formula (1) follow from? It is necessary to give relevant links or justify the transformation formula (1).
Response: Thanks for your advice. The transformation formula (1) follows from the reference [34] but with the different value of z0. We have modified the value of current z0 to the one described in ref. [34] as z0=1.52×10-4, and the reference has been added in the revised manuscript.
[34] Lu Y.; Zhang B.; Perrie W.; Mouche A. A.; Li X.; Wang H. A C-band Geophysical Model Function for Determining Coastal Wind Speed Using Synthetic Aperture Radar[J]. IEEE J. Sel. Top. Appl. Earth Observ. Remote Sens. 2018, 11, 2417-2428.
2) For what reasons is z0=… chosen? Justifications are required.
Response: The constant z0 represents the roughness length. The value of z0 depends on the nature of the terrain and must be determined empirically, and it is sometimes taken as 15% of the average height of the roughness elements [35]. Hence, the typical value is 1.52Í10-4 [18, 34, 35]. The above words, after modification, have been added in Line 133-135 of the revised manuscript.
[35] Peixoto J. P.; Oort A. H.; Lorenz E. N. Physics of climate[M]. New York: American Institute of Physics, 1992.
3) Missing formula (2).
Response: Many thanks for pointing out the error. We have adjusted the sequence number of the formulas in the revised manuscript.
4) The text is a bit boring.
Response: We have thoroughly revised the manuscript, and the quality of the language and writing has also been considerably improved. Please see the revised version and the tracking changed version for details.

Reviewer 2 Report
Dear Authors,
Your manuscript entitled “ Assessment of Sea Surface Wind Retrieval from C-band Miniaturized SAR imagery” is scientifically interesting and systematic. It also brings new and valuable ideas and results to the satellite monitoring of sea surface wind.
The manuscript can be published in Sensors. Sincerely yours,
Reviewer
Author Response
Reviewer 2: Your manuscript entitled “Assessment of Sea Surface Wind Retrieval from C-band Miniaturized SAR imagery” is scientifically interesting and systematic. It also brings new and valuable ideas and results to the satellite monitoring of sea surface wind. The manuscript can be published in Sensors.
Response: Many thanks for your positive comments. We appreciate that you found this manuscript is scientifically interesting and systematic.

Reviewer 3 Report
This study reports on the measurement of sea surface winds using a compact SAR system.
Miniaturization of the system is useful.
However, this report fails to demonstrate its usefulness.
(1) Equation 5 uses the sea surface winds from the CCMP.
Therefore, it is obvious that the estimated wind is somewhat close to the CCMP winds.
The CCMP estimates wind speeds with sufficient accuracy.
If that is the case, is it not necessary to use data from this SAR system?
The spatial resolution of SAR wind may be high.
However, since the system uses CCMP sea winds, the data is smoothed with respect to space, so in effect, the spatial resolution would not be as high as the CCMP winds.
(2) What about wind direction?
(3) You say this system costs less and weighs less (185 kg), but how different is it from any previous system?
Author Response
Reviewer 3: This study reports on the measurement of sea surface winds using a compact SAR system. Miniaturization of the system is useful. However, this report fails to demonstrate its usefulness.
Response: We are grateful for your detailed review of this manuscript. Please find below our responses to your comments and suggestions.
(1) Equation 5 uses the sea surface winds from the CCMP. Therefore, it is obvious that the estimated wind is somewhat close to the CCMP winds. The CCMP estimates wind speeds with sufficient accuracy. If that is the case, is it not necessary to use data from this SAR system? The spatial resolution of SAR wind may be high. However, since the system uses CCMP sea winds, the data is smoothed with respect to space, so in effect, the spatial resolution would not be as high as the CCMP winds.
Response: The spatial resolution of CCMP is 0.25°Í0.25° (approximately 25km × 25km). This implies that wind speed value at each grid point reflects the average wind speed within a 25km × 25km area centered around that grid point. However, wind speed and sea surface roughness exhibit non-uniform characteristics within this area, particularly in coastal regions where notable variations are observed. Therefore, it is necessary for retrieving high-resolution wind fields from SAR.
In 2002, Portabella et al. proposed a variational method for wind field retrieval based on Bayesian theory [27]. This method combines NRCS (Normalized Radar Cross Section), geophysical model function, and external wind field data to construct a variational equation (cost function). The optimal wind vector is determined by seeking the minimum value of the variational equation. Additionally, the external wind data are not limited to CCMP and can include ECMWF data [28, 29] and Numerical Weather Prediction data [27, 30]. To ensure independent validation, the wind field data used for computing the minimum cost function should not be employed for evaluating the retrieved wind field.
As mentioned in Line 226–229 of page 5, the SAR imagery is partitioned into multiple wind cells (sub-SAR images), each possessing a coverage of 1 km × 1 km. Hence, the spatial resolution of retrieved SSWF is 1 km × 1 km. However, in comparison with ERA5, we averaged the SSWFs from each entire SAR image and compared it with the SSWF at the ERA5 grid point included in the SAR image.
For clarity, we have added “Hence, the spatial resolution of retrieved SSWF is 1 km × 1 km.” to Line 227–228 in the revised manuscript.
[28] Ren L.; Yang J.; Mouche A.; Wang H.; Zheng G.; Wang J.; Zhang H.; Lou X. Chen P. Assessments of ocean wind retrieval schemes used for Chinese Gaofen-3 synthetic aperture radar co-polarized data[J]. IEEE Trans. Geosci. Remote Sens. 2019, 57, 7075-7085.
[29] Wang H.; Yang J.; Mouche A.; Shao W.; Zhu J.; Ren L.; Xie C. GF-3 SAR ocean wind retrieval: The first view and preliminary assessment[J]. Remote Sens. 2017, 9, 694.
[30] Mouche A. Sentinel-1 ocean wind fields (OWI) algorithm definition[J]. Sentinel-1 IPF Reference:(S1-TN-CLS-52-9049) Report, 2011, 1-75.
(2) What about wind direction?
Response: Our work currently focuses on the assessment of retrieved wind speed, while the retrieval of wind direction has not yet been completed. Additionally, in ocean dynamics research, the impact of wind speed is much greater than that of wind direction. Therefore, this study is more concerned with wind speed rather than wind direction.
(3) You say this system costs less and weighs less (185 kg), but how different is it from any previous system?
Response: In contrast to previous SAR systems, HiSea-1 and Chaohu-1 exhibit comparable imaging performance despite their reduced size and weight. As a comparison, the Gaofen-3 SAR satellite’s weight is 2750 kg and its cost of power is 15 kW, while HiSea-1 SAR satellite’s weight is 185 kg and its cost of power is 2.5 kW [19]. In a word, this study aimed at investigating the potential of the two miniaturized SAR in SSWF retrieval. For clarity, the above words have been added in Line 375–381 in the revised manuscript.

Round 2
Reviewer 3 Report
The authors have not responded properly to the previous comments (1) and (2).
I recommend rejecting them once to allow sufficient revision time.
(1) It is known that the spatial resolution of the NRCS of SAR is higher than that of the CCMP data.
They also use CCMP winds for the cost function; since CCMP wind speeds agree with buoy wind speeds, it is not surprising that SAR wind speeds agree with buoy wind speeds.
However, the advantage of SAR wind speed is not shown.
The paper does not respond to the comment, "Since the system uses CCMP sea winds, the data is smoothed with respect to space, so in effect, the spatial resolution would not be as high as the CCMP winds."
To demonstrate high spatial resolution of SAR winds, the case of a passing atmospheric front, etc. should be illustrated instead of the case of nearly unidirectional wind distribution shown in this paper.
With 10 months of data period, there should be such a case.
(2) It is only strange that Equation 4 requires wind vectors but does not compare wind directions.
The response that the retrieval of wind direction has not yet been completed is clearly wrong.
Comparison of wind direction is also necessary.
Author Response
The authors have not responded properly to the previous comments (1) and (2).I recommend rejecting them once to allow sufficient revision time.
Response: Thank you for reviewing our manuscript and providing valuable feedback. We apologize for not adequately addressing your previous comments, here we try to answer them more clearly.
(1) It is known that the spatial resolution of the NRCS of SAR is higher than that of the CCMP data. They also use CCMP winds for the cost function; since CCMP wind speeds agree with buoy wind speeds, it is not surprising that SAR wind speeds agree with buoy wind speeds. However, the advantage of SAR wind speed is not shown. The paper does not respond to the comment, "Since the system uses CCMP sea winds, the data is smoothed with respect to space, so in effect, the spatial resolution would not be as high as the CCMP winds."
Response: In this manuscript, we adopt the variational method to inverse the SSWFs. In 2002, the method was firstly described by M. Portabella, A. Stoffelen, and J. A. Johannessen in Reference [27]. Then it has been widely used for SSWFs production of Sentinel-1 and Gaofen-3 SAR satellites [28-30].
The idea of this method to find the optimal wind vector by a cost function and optimization process. Since the relation between SSWFs and SAR observations is highly nonlinear, the method combines high-resolution SAR backscatter intensities (σ0), geophysical model function (GMF), and external coarse-resolution wind field data to retrieve the most probable wind vector, assuming that all sources of information contain errors and that these are well characterized, including their spatial correlation. Besides CCMP, coarse-resolution wind information of Numerical Weather Prediction (NWP) has also been used as the external wind data. Although their resolution is lower compared to SAR, but the optimization process can find a best wind vector as the SAR σ0 contains high-resolution spatial information of local wind.
Here the SSWF retrieval process used in this study is similar to the one described in Reference [30], with the only difference being the choice of external wind field data (CCMP) instead of NWP data. Both References [29] and [30] segmented SAR images into 1 km × 1 km sub-images for wind field retrieval, while Reference [28] used 5 km × 5 km image segments for wind field retrieval. In Reference [28], the retrieved wind speeds were compared to ASCAT, HY2A-SCAT, and buoy data, demonstrating higher accuracy compared to the direct retrieval method (the direct retrieval method was described in Line 77-86).
To demonstrate high spatial resolution of SAR winds, the case of a passing atmospheric front, etc. should be illustrated instead of the case of nearly unidirectional wind distribution shown in this paper. With 10 months of data period, there should be such a case.
Response: We have addressed your comment by adding an example in the revised manuscript to demonstrate the high spatial resolution of SAR winds. This example has been included in Figure 3, replacing the previous Figure 3b and 3c. Additionally, we added the Figure 3e to display the changing wind directions in SAR images.
(2) It is only strange that Equation 4 requires wind vectors but does not compare wind directions. The response that the retrieval of wind direction has not yet been completed is clearly wrong. Comparison of wind direction is also necessary.
Response: In our revised manuscript, we have incorporated the comparison of wind directions and provided the corresponding analysis. The figures have been added to Figure 5.